# Implicit Calibration Using Probable Fixation Targets

**DOI:** 10.3390/s19010216

**Published:** 2019-01-08

**Authors:** Pawel Kasprowski, Katarzyna Harȩżlak, Przemysław Skurowski

**Affiliations:** Institute of Informatics, Silesian University of Technology, Akademicka 16, 44-100 Gliwice, Poland; katarzyna.harezlak@polsl.pl (K.H.); przemyslaw.skurowski@polsl.pl (P.S.)

**Keywords:** eye tracking, calibration, eye movement, optimization

## Abstract

Proper calibration of eye movement signal registered by an eye tracker seems to be one of the main challenges in popularizing eye trackers as yet another user-input device. Classic calibration methods taking time and imposing unnatural behavior on eyes must be replaced by intelligent methods that are able to calibrate the signal without conscious cooperation by the user. Such an implicit calibration requires some knowledge about the stimulus a user is looking at and takes into account this information to predict probable gaze targets. This paper describes a possible method to perform implicit calibration: it starts with finding probable fixation targets (PFTs), then it uses these targets to build a mapping-probable gaze path. Various algorithms that may be used for finding PFTs and mappings are presented in the paper and errors are calculated using two datasets registered with two different types of eye trackers. The results show that although for now the implicit calibration provides results worse than the classic one, it may be comparable with it and sufficient for some applications.

## 1. Introduction

With growing access to cheap eye-tracking devices using simple web cameras, and advancements in image processing and analysis, it has become possible to incorporate eye tracking as yet another human–computer interface.

The accuracy and precision of cheap eye trackers (priced less than $500) are lower than the high-end eye trackers (for more than $5000); however, there are numerous examples that such eye trackers may be used not only for simple tasks but also for research [1]. There are publications confirming that e.g., The Eye Tribe [2], Gazepoint GP3 [3] or Tobii EyeX [4] may give reliable eye-tracking results when certain conditions are met.

However, one of the most important problems when considering eye-tracking usage on an end-user market is the necessity of calibration. Most eye trackers require to be calibrated prior to each usage. Unfortunately, the calibration procedures currently used are cumbersome and inconvenient. Additionally, to achieve reliable results, they often require qualified operators that control the process. As a result, eye tracking is reliable only in laboratory conditions and usage of eye tracking in the wild by unqualified users is often impossible.

The goal of the research summarized in this paper is to develop calibration methods that make using eye tracking possible also for unattended and inexperienced end-users. The idea that the calibration may be done implicitly, during normal user’s activities is our main research hypothesis. The possibility of such a calibration has been confirmed in several papers [5,6,7,8]; however, the main challenge is to achieve generality of the solution and accuracy that is sufficient for end-user eye-tracking usage in various eye-tracking applications.

### 1.1. Calibration

Every eye tracker produces some output, which may be multidimensional and depends on the type of the eye tracker used. It may be information regarding image coordinates of the pupil center, light reflections (glints) positions, signal intensity (for Direct Infra Red Oculography or Electrooculography ) or just the whole eye image used by appearance-based methods. This output may be treated as a set of attributes $e_{1},\ldots \;e_{n}$. The purpose of the calibration is to build a model (calibration function $f_{calibration}$) that converts this output into information about the gaze point—a place where the examined subject is looking at a particular moment. The gaze point is given in scene coordinates and typically is presented as a point in a two-dimensional scene space (Equation (Equation 1)) but may also be defined in three-dimensional (3D) space [9,10]. The scene coordinates may be specified in relation to the observed screen (in the case of eye trackers working with screens) or in relation to scene camera coordinates (in the case of eye trackers equipped with a scene camera) [11].
(1)$$gaze_{i}\left(x,y\right)=f_{calibration}\left(e_{1},e_{2},\ldots e_{n}\right)$$

During the classic calibration procedure, the subject is expected to look for some time at predefined locations in the scene (so called targets). When a certain degree of eye tracker output data with the corresponding known gaze locations is available, it is possible to build the calibration function that will be able to map a future eye tracker output to gaze points. The quality of the function depends on the amount of training data available, diversity of recorded gaze points’ positions and quality of the recordings [11,12]. While the two first aspects may be adjusted by prolonging the calibration procedure, the latter is difficult to inspect.

Such a calibration scenario typically requires focusing on one target and leaving the eyes stable for a long time to collect the necessary data. As human eyes tend to change focus several times a second, gazing at a static target is not a natural task. Therefore, such a procedure is cumbersome and inconvenient for users and prone to errors when participants do not look where they are expected to because of tiredness or loss of focus.

Another problem is that the accuracy of results derived by the calibration function tends to decrease with time [5,13]. To solve this problem, the calibration procedure must be repeated from time to time which makes the usage of an eye tracker even more inconvenient.

These issues have resulted in the search for alternative calibration methods. There were many other scenarios tested, including: a click-based calibration [14], smooth pursuit calibration [15], vestibulo-ocular reflex (VOR)-based calibration [16] and others. However, all these methods share the same burden: they take time, are not very convenient for users and their results lose accuracy with time. For this reason, there is a growing interest in methods able to perform an implicit calibration, during a normal user’s activities.

### 1.2. Implicit Calibration

The main assumption of the presented research is that when information about an observed scene is available, it is possible to predict—with some probability—at what location a subject will look. Having this information and data from an eye-tracking device, it is possible to construct a model that can estimate the gaze point, similarly to the classic calibration. Such a calibration does not require any explicit nor active cooperation of the user—in fact, the users may even not be aware that they are being calibrated. Therefore, such a calibration is called implicit calibration and because it may be performed online, during the whole eye-tracking session it may be also treated as continuous calibration.

Implicit calibration always requires information about the scene, which is explored by the user. The probability that the user looks at specific points may be calculated using various criteria. The simplest algorithm may use the center location paradigm that states that most of the time the user is looking at the center of a scene [17]. More sophisticated algorithms may use saliency map models, or any other predictions [18]. It is worth noting that the implicit calibration may replace a classic calibration but may also be used only to improve or maintain an already performed calibration [19,20].

### 1.3. The State of the Art

The problem of implicit calibration has been already analyzed in several publications; however, to the best of our knowledge, there are still no out-of-the-box solutions, which may be directly used by potential users.

One of the first papers using solutions similar to implicit calibration was [5]. The authors aimed to correct systematic calibration drift using clues regarding gaze position obtained during a participant’s work. The clues were locations of mouse clicks (with the assumption that people look where they click) and were used to check the calibration quality and invoke recalibration if necessary. A similar assumption was made in [20], where the initial calibration was completely removed, and the calibration was done only based on mouse-click positions.

When the number of clicks is insufficient, it is possible to use other information about the scene, e.g., identify objects that should be fixated to accomplish tasks, such as labels, icons to find, etc. This idea was used to correct drift errors in the calibrated signal in [13,19].

When a scene does not consist of easily distinguishable objects, as in the case of user interfaces, a more general solution must be used that tries to predict fixation targets taking into account only information about the scene a user is looking at. Such a solution involves creation of a saliency map—a function that estimates for each scene point a probability that user will look at this point [21].

Probably the first paper in which saliency maps were used as an input to the calibration model was [7], extended later in [22]. The authors used graph-based visual saliency (GBVS) model [23] and face detector to calculate saliency for each frame of a movie. The saliency map was later aggregated to improve optimization. The same saliency model was used in [17]. Another example includes [8] where the map was built using a self-developed algorithm based on the Regression Convolutional Neural Network (RCNN). Instead of using algorithmic saliency model it is also possible to use gaze data recorded earlier for the same image as it was done in [24]. However, such a solution is possible only when the same stimulus is presented many times and is not feasible in general case.

Some methods use not only a pure saliency maps but use also natural constraints like center bias that assumes that gaze locations are biased towards the center of the scene during free viewing [25,26].

When saliency for every frame is known, it may be used to estimate parameters of a calibration model that, for the given data, maximizes the likelihood that the computed gaze falls in the region of high saliency. There are different optimization methods used: Gaussian process regression [22], Bayesian network [17], K-closest points and mixture model methods [24] or minimization of the Kullback-Leibler divergence between distributions [8].

Most experiments published so far recorded a set of gazes while a person was watching static images [8,17,24,27]. Only in [22] the authors used a set of frames from a movie and computed a saliency map for each frame. Results presented in one of the recent papers [28] showed that implicit calibration may be used also in an unconstrained environment.

### 1.4. Paper’s Contribution

This paper describes a novel technique to perform an implicit calibration. This technique is based on the assumption that in most cases there are only a few points (targets) in a scene being observed where users may fixate their gazes. After finding these targets it will be possible to build a calibration model (Figure 1). As there may be more than one target for each fixation, such targets are named probable fixation targets (PFT).

This paper discusses all consecutive steps that are necessary to perform the implicit calibration using PFTs. The steps include:saliency maps prediction,conversion of saliency maps to lists of PFTs,choosing a mapping (a set of PFTs) using miscellaneous criteria and heuristic algorithms,using the chosen PFT mapping to build a calibration model.

The experimental part of the paper checks the usability of the proposed solutions using two datasets containing uncalibrated eye movement data recorded by different eye-tracking devices and various stimulations. The datasets are publicly available and may serve as a convenient testbed for future calibration algorithms.

The paper summarizes the idea introduced in two conference publications [6,29]; however, it significantly extends it by:Analyses of algorithms for PFT (targets) creation using saliency maps.Introduction of new mapping comparison functions.More extensive analyses of results and possible algorithms’ parameters.Introduction of two new datasets.Implementation of all presented algorithms in a publicly available new version of the ETCAL library which has been created for calibration purposes and is available at GitHub [30].

## 2. Methods and Materials

The original idea, used during the research presented in this paper, is introducing an additional step of processing during which a saliency map is used to build a short list of PFTs. Having a limited number of such targets for every fixation, it is possible to build the model assuming that most of the time a user would look at one of these targets.

All processing steps of the algorithm are presented in Figure 2. It gets two input signals: information about a scene being observed and eye tracker output. The eye tracker output is a set of attributes ($e_{1},e_{2},\ldots ,e_{n}$) and it will be denoted as $E_{i}\left(e\right)$ in the subsequent text. The *e* may be any set of attributes depending on the type of eye tracker used: eye center coordinates in the eye camera, eye-glint vector, electrooculogram, etc. The first step is the extraction of PFTs from scene information (Targets Extractor) and fixations from an eye tracker output (Fixations Extractor). Then the list of PFTs together with the corresponding fixations $F_{i}\left(e\right)$ is used to build the training set (Training Dataset Builder). The training set consists of *M* fixations and for each fixation $F_{i}$ there is a list of $m_{i}$ PFTs: {$T_{i}^{\left(1\right)},T_{i}^{\left(2\right)},\ldots ,T_{i}^{\left(m_{i}\right)}$} where each *T* is a (x,y) pair (see Equation (Equation 2)).
(2)$$\begin{array}{c} F_{1}:E_{1}\left(e\right),\left\{T_{1}^{\left(1\right)},T_{1}^{\left(2\right)},\ldots ,T_{1}^{\left(m_{1}\right)}\right\},\\ F_{2}:E_{2}\left(e\right),\left\{T_{2}^{\left(1\right)},T_{2}^{\left(2\right)},\ldots ,T_{2}^{\left(m_{2}\right)}\right\},\\ \ldots \\ F_{M}:E_{M}\left(e\right),\left\{T_{M}^{\left(1\right)},T_{M}^{\left(2\right)},\ldots ,T_{M}^{\left(m_{M}\right)}\right\}. \end{array}$$

The training set is analyzed to find the possible targets (Target Mapper). The mapper chooses one PFT for each fixation (see Figure 3), so then it is possible to build the calibration model (Calibration Model Builder). When the calibration model is built, it may be used to calibrate subsequent eye tracker output (Calibrator).

It is important to note that the whole process may be iterative—new output together with new targets may be continuously used to improve the calibration model.

Every step listed in this section is described in detail in the following subsections. Specifically:Extraction of targets (PFTs) from saliency information.Choosing the mappings comparison function (MCF).Developing heuristic algorithms for searching the mapping space (genetic and incremental)Building a calibration model using the ETCAL library ([30]).

### 2.1. Extraction of Targets (PFTs) from Saliency Information

The main requirement for the implicit calibration is knowledge of the observed scene. Based on its analysis it is possible to build a function calculating, for each point of the scene, the probability that a subject will focus on it.

The simplest algorithm may use center location paradigm that states that most of the time user is looking at the center of a scene [17]. More sophisticated algorithms may use various predictions [18]; however, the most popular way to calculate the probability function is the application of a saliency model that produces a saliency map [31].

Saliency maps are computational models of human visual attention, the concept was conceived by Koch and Ullman in [32]. It is a topographical map of a scene that describes scene fragments as weights—how likely a human gaze will be fixed at a certain location. However, despite apparent similarity, saliency maps should not be interpreted or confused with the probability density function of the subconscious process. Contrary to random processes, selection of fixations is not a stochastic process of independent events—it is chaotic [33]—therefore describing it as PDF would be irrelevant.

The first practical, biologically inspired, implementation of the saliency model was proposed by Itti, Koch and Niebur [21]. It started intense research on further models based on various fundamental premises and assumptions [34]. It may be the bottom-up approach that uses only image parameters (like contrast) or top-down that uses knowledge regarding human behavior (like a face detector). Recently, methods that combine both approaches have gained popularity—e.g., the model by Le Meur [35] modelling a saliency map with the low-level visual feature-based saliency map and modelling the fixation selection based on: context-dependent statistics of saccade length and angle, inhibition of return, and forgetting of previous fixations.

There were nine different saliency models used in this research. The following list enumerates and briefly describes each of the employed saliency map models:BMS—Boolean maps saliency [36]—a graphical model based on a series of thresholds and sophisticated mathematical morphology,CovSal [37]—based on the innovation introduced in a certain area when compared to its neighborhood in image, using covariance as a measure,FES [38]—Bayesian bi-variate modelling of saliency based on the center-surround model at multiple scales.GBVS—the graph-based visual saliency model by Harel et al. [23], location vectors dissimilarity used as graph edge values,IttiKoch—a biologically plausible model, based on modelling of the underlying processes [21],LDS—Learning Discriminative Subspaces [39]—based on the image segments projection onto pre-trained dictionary patches of PCA reduced visual features.RARE2012 [40]—based on the multiscale innovation (rarity) in location, identified with Gabor filters and information theoretic measure self-information.SR-PQFT—a spectral residual (SR) model (as proposed by Hou and Zhang in [41]), but in a variant that was developed by Schauerte and Stiefelhagen [42] for color images, where color values are quaternion encoded in opponent color space and the quaternion formula of Fourier transform is employed (PQFT),SWD [43]—spatially weighted dissimilarity—saliency model also based on the difference from the surrounding area, where outlying segments are identified in PCA reduced feature space.

Apart from artificial saliency models, we also created two sets of targets based on human observation:GT—users’ fixation maps created using eye-tracking information gathered during genuine observations undertaken by several participants.EXP—additionally, we also tested if PFTs may be set manually, by the ‘fixation expert’. One of the co-authors of the paper manually defined PFTs for each image, based on his experience (before seeing any saliency map).

Every saliency map had to be transformed to a set of PFT. The list should contain only a limited number of points with high saliency. For this purpose, each saliency map was processed using the following cascade of transformations from the OpenCV [44] library:histogram equalizationbinarization with a threshold equal to 240series of dilations and erosionsedge detection using the Canny algorithm [45]contours extraction using the Suzuki algorithm [46]

Then, for every contour identified, the center of the mass was computed and added as a probable fixation target. The number of PFTs varied for images and maps and ranged from 1 to 16. Of course, this approach is just one of the possibilities and better algorithms converting saliency maps into targets may be considered as an element of future studies.

Examples of maps and calculated PFTs are presented in Figure 4.

### 2.2. Choosing the Best Mappings Comparison Function (MCF)

To calculate parameters of a calibration function it is necessary to choose one target with index $c_{i}$ for every fixation $F_{i}$. Therefore, for every fixation an index of one of its PFTs is chosen denoted as $c_{i}$ where $c_{i}\in \langle 1,m_{i}\rangle$. The list of chosen indexes for all fixations in the training set $c_{1}\ldots c_{M}$ is called a *mapping*.
(3)$$mapping=\{c_{1},c_{2},\ldots ,c_{M}\}$$

The question is how to find a correct mapping from a list of all possible mappings. Without ground-truth information the only data that may be used is the uncalibrated output from an eye tracker. It is necessary to define criteria that may be used to compare two distinct mappings. Such a criteria function is called mappings comparison function (MCF).

#### 2.2.1. Regression Based Mappings Comparison Function

One of the possible solutions is to evaluate the specific mapping by: (1) building a linear function that recalculates eye tracker output to target coordinates $\left(g_{x},g_{y}\right)$ and (2) checking the function error for the same target points.

The linear function that recalculates eye tracker output separately to vertical and horizontal coordinates of the gaze may be defined as follows:(4)$$\begin{array}{c} g_{x}=\beta_{0x}+\beta_{1x}\cdot e_{1}+\beta_{1x}\cdot e_{2}+\ldots +\beta_{nx}\cdot e_{n} \end{array}$$(5)$$\begin{array}{c} g_{y}=\beta_{0y}+\beta_{1y}\cdot e_{1}+\beta_{1y}\cdot e_{2}+\ldots +\beta_{ny}\cdot e_{n} \end{array}$$
where $g_{x}$ and $g_{y}$ are horizontal and vertical gaze coordinates respectively and parameters $\beta_{ij}$ are regression coefficients calculated using the Levenberg-Marquardt algorithm based on known eye tracker output $E_{i}\left(e_{1}\;\ldots \;e_{n}\right)$ and the corresponding PFT values $T_{i}\left(t_{x},t_{y}\right)$.

The next step is the calculation of the coefficient of determination $R^{2}$, which compares a function’s results with the target values:(6)$$R_{x}^{2}\left(t_{x},g_{x}\right)=1-\frac{\sum_{i}^{M}{\left(g_{x}\left(i\right)-t_{x}\left(i\right)\right)}^{2}}{\sum_{i}^{M}{\left(t_{x}\left(i\right)-\bar{t_{x}}\right)}^{2}}$$
where *M* is the number of fixations and $\bar{t_{x}}$ is the average of all reference values $t_{x}\left(i\right)$. $R_{x}^{2}$ is equal to 1 when the model fits perfectly, i.e., every $g_{x}\left(i\right)$ point is exactly at the same location as the corresponding $t_{x}\left(i\right)$ point.

The same formula may be used for the vertical axis to calculate $R_{y}^{2}\left(t_{y},g_{y}\right)$. Both results are then combined together to form one value that is treated as the ‘quality’ of the mapping:(7)$$MCF_{REG}\left(mapping\right)={\left(R_{x}^{2}\left(t_{x},g_{x}\right)\cdot R_{y}^{2}\left(t_{y},g_{y}\right)\right)}^{2}$$

It may be assumed that the higher value of $MCF_{REG}$ means that it was easier to build a linear function recalculating the eye tracker output to the given mapping and therefore the probability that this mapping is correct (i.e., a person really looked at these targets) is higher.

#### 2.2.2. Mapping Comparison Function Based on Relative Distances

The $MCF_{REG}$ function is applicable in any general case; however, it is not very efficient as it requires calculation of a regression model and its error for each mapping. Another option is to use the fact that some information concerning a spatial correlation between gazes may exist even in an uncalibrated signal.

For instance, we can assume that if we calculate the Euclidean distance between eye tracker data points ($E_{i}$) it should be correlated with distances between gaze points ($T_{i}$) corresponding with those outputs. Therefore, when $∥E_{i},E_{j}∥<∥E_{j},E_{k}∥$, we can expect that for the chosen targets, the distance $∥T_{i},T_{j}∥$ will also be lower than $∥T_{j},T_{k}∥$.

This assumption was used to calculate another MCF. For each three fixations i,j,k the proportion of distances was calculated using the Formula (Equation 8):
(8)$$P_{ijk}=\left(\frac{∥E_{i},E_{j}∥}{∥E_{i},E_{k}∥}-1\right)/\left(\frac{∥T_{i},T_{j}∥}{∥T_{i},T_{k}∥}-1\right)$$
and then the *P* values for all triples in the mapping were used to calculate the value of $MCF_{DIST}$ for the mapping:(9)$$MCF_{DIST}\left(mapping\right)=\sum_{i=1}^{n-2}\sum_{j=i+1}^{n-1}\sum_{k=j+1}^{n}\left\{\begin{array}{cc} 1, & P_{ijk}>0\\ 0, & otherwise \end{array}\right.$$

#### 2.2.3. MCF Based on Correlation between Distances

The MCF defined in the previous section is time consuming as it requires comparison of every triple of fixations. Therefore, a faster method was also proposed and tested that works based on the same principle (distances comparison) but is less complex.

At first the distances are calculated for each neighboring pair $\left(E_{i},E_{i+1}\right)$ to build a variable ∥E∥. Then all neighboring distances $\left(T_{i},T_{i+1}\right)$ are used to form another variable ∥T∥. We assume that relations between distances in both variables will be similar, therefore the final value of $MCF_{DISTC}$ is calculated as a correlation coefficient between both variables:(10)$$MCF_{DISTC}\left(mapping\right)=\rho \left(∥E∥,∥T∥\right)$$

Once again, the assumption is that a higher MCF value (i.e., higher correlation between distances) means a better mapping.

#### 2.2.4. MCF Based on Direction

Another assumption taken into account was that the shape of the eye tracker output signal should be similar to the resulting gaze path. For instance, when the head-mounted eye tracker with eye camera is used, and an output signal consists of eye center coordinates as visible in the eye camera, it may be assumed that a correlation exists between the direction of eye center changes and gaze positions. Generally, when the eye center moves down in the camera image, it may be expected that the gaze should move down as well. Similarly, when the eye center moves right, it may be expected that the gaze position moves left (due to the mirror effect) and so on. Therefore, the shape of eye center points should be similar to the shape of gaze points (see Figure 5).

This assumption may always be used when the eye tracker produces only two attributes. It seems to be a problematic constraint; however, it occurs that many eye trackers fulfill this condition. For example head-worn eye trackers that output eye center coordinates, remote eye trackers that output eye center–glint vector, EOG eye trackers that produce two voltage signals or Direct Infra Red eye trackers such as Jazz-Novo [12] that output luminance information.

Taking this assumption into account it was possible to define another MCF to evaluate a mapping. At first, for two fixations $F_{i}$ and $F_{j}$ it is possible to calculate a vector between eye center positions for these fixations:(11)$$\vec{E_{ij}}=\left(e_{x}\left(j\right)-e_{x}\left(i\right),e_{y}\left(j\right)-e_{y}\left(i\right)\right)$$

In the case of eye camera and eye center coordinates the vector must be flipped horizontally:(12)$$\vec{E_{ij}}=\left(e_{x}\left(i\right)-e_{x}\left(j\right),e_{y}\left(j\right)-e_{y}\left(i\right)\right)$$

It is also possible to calculate a vector between two targets:(13)$$\vec{T_{ij}}=\left(t_{x}\left(j\right)-t_{x}\left(i\right),t_{y}\left(j\right)-t_{y}\left(i\right)\right)$$

Due to the supposed similarity between eye centers and gaze points it may be assumed that both vectors should be more or less parallel to each other (see Figure 5).

When a cosine of angle between $\vec{E_{ij}}$ and $\vec{T_{ij}}$ vectors is calculated for every two fixations, it may be used to establish a MCF:(14)$$MCF_{DIR}\left(mapping\right)=\sum_{i=1}^{n-1}\sum_{j=i+1}^{n}cos(\vec{E_{ij}},\vec{T_{ij}})$$

For this MCF it is assumed that the mapping with a higher value should yield better calibration results (see Figure 5 for visual explanation).

#### 2.2.5. Fusion of More than One MCF

The MCF presented in the previous sections should give some information about the mapping’s quality. It is also possible to combine several MCFs with different weights to create a new—possibly more robust—mapping comparison function according to formula (Equation (Equation 15)):(15)$$MCF_{FUS}\left(mapping\right)=w_{0}\times MCF_{i}+w_{1}\times MCF_{j}+\ldots$$

Such a function with correctly tuned weights may give better results than each of the MCFs separately.

### 2.3. Heuristic Algorithms for Searching the Mapping Space

One problem is to find the MCF, which enables comparison between mappings, another problem is to find the best mapping according to the given MCF. This issue is not trivial. If there are *n* fixations and $m_{i}$ targets for every fixation the number of possible mappings is $m_{1}\times m_{2}\times \ldots \times m_{n}$, so it is not feasible to do an exhaustive search through all mappings. Therefore, a heuristic algorithm must be used. Two possible heuristics are proposed in this paper: genetic and incremental.

#### 2.3.1. Genetic Algorithm

The genetic algorithm treats each target as a gene and each mapping as a chromosome (set of genes). The task is to find the best possible chromosome. At first the algorithm randomly chooses some number of mappings (e.g., 200). Then it evaluates every mapping and mixes some of the best of them to build new mappings (chromosomes) using cross overs. Additionally, each chromosome is mutated—some number of genes (targets) is changed randomly. The process is conducted in many iterations (e.g., 1000) to find the best possible mappings. The algorithm is described in detail in Algorithm 1.
**Algorithm 1** Heuristic genetic algorithm searching for the best mapping**procedure**FindBestMappping    pop←200randommappings    iterations=0    noProgress=0    **while**
iterations<1000 and noProgress<200
**do**        newPop←∅        newPop.add(pop.bestMapping)        **for**
i=2→200
**do**           mapping1=chooseGoodRandomMapping(pop)           mapping2=chooseGoodRandomMapping(pop)           newMapping=crossOver(mapping1,mapping2)           newPop.add(newMapping)        **for**
i=2→200
**do**           mutate(newPop.mapping(i))        **if**
newPop.bestMapping>pop.bestMapping
**then**           noProgress=0        **else**           noProgress++        pop=newPop    **return**
pop.bestMapping**procedure**chooseGoodRandomMapping    Choose 5 random mappings and return the one with the best fitness**procedure**crossOver(mapping1,mapping2)    Return a new mapping with values randomly chosen from the first or the second mapping**procedure**mutate(mapping)    Randomly change 1.5% indexes in the mapping 

#### 2.3.2. Incremental Mapping

The genetic method explained in the previous section has two significant disadvantages: it needs all data to start optimization and it is quite slow. To use the implicit calibration in a real-time scenario it is necessary to propose an algorithm that can calculate the calibration model adaptively and improve it when new data arrives.

The algorithm for incremental implicit calibration works as follows: it starts when only a few fixations are available and creates a list of all possible mappings. When the next fixation (with its own PFTs) arrives, the algorithm creates a new list by adding one of the new PFT indexes to every mapping from the previous list (see Algorithm 2). Every new mapping is evaluated according to the score calculated for the chosen optimization criterion (Line 14). The list of mappings is sorted in descending order (Line 6). The same is done for every new fixation—when fixation n+1 is added and there are *x* mappings (of length *n*) on the list, a new list is created with $x\times m_{n+1}$ mappings. The number of indexes in every mapping on the new list is n+1. The mappings are sorted in descending order according to the given criterion. When the number of mappings on the list exceeds a predefined maximal number cutoff (e.g., 100), the algorithm removes mappings with indexes above cutoff (the ones with the lowest criterion value) (Line 7).
**Algorithm 2** Iterative algorithm finding cutoff best mappings after adding a new fixation with newTargetIndexes to mappings created so far  1:**procedure**NewMappings(oldMappings,newTargetIndexes)  2:    newMappings←∅  3:    **for all**
mapping∈oldMappings and index∈newTargetIndexes
**do**  4:        newMapping←AddTarget(mapping,index)  5:        newMappings.add(newMapping)  6:    SortDescending(newMappings)  7:    RemoveElementsOverCutoff(newMappings,cutoff)  8:    **return**
newMappings  9:**procedure**AddTarget(oldMapping,newIndex)10:    newMapping←∅11:    **for all**
index∈oldMapping
**do**12:        newMapping.add(index)13:    newMapping.add(newIndex)14:    newMapping.calcValue()15:    **return**
newMapping 

A procedure outcome may be used at every moment and applied to build a calibration model at the early stage of data processing, even after several steps. It may be expected that the calibration error will be high at the beginning; however, it will decrease with the number of fixations used.

The main advantage of this solution, as compared to heuristic algorithms is that it is considerably faster—the complexity is linear because calculation of a value for a new mapping may use the already calculated value of the mapping one index shorter. Moreover, as it was mentioned above, the results are available from the beginning of the experiment (although are only really valuable after some time—see the Results section for details).

### 2.4. Removing Irrelevant Fixations

When the best mapping is chosen using the steps explained in the previous sections, it should be additionally processed to remove irrelevant fixations. The rationale behind this step is that even when participants look at the predefined PFTs, there will always be some number of fixations that fall somewhere else.

At first all fixations—pairs $\left\{E_{i}\left(e_{1},\ldots \;e_{n}\right),T_{i}^{\left(c_{i}\right)}\left(x,y\right)\right\}$—are used to calculate parameters of the linear polynomial calibration function $f_{l}$ using the Levenberg-Marquardt (LM) damped least-squares algorithm [47]. In the next step gaze coordinates are calculated for every pair based on the following function:(16)$$gaze_{i}\left(x,y\right)=f_{l}\left(E_{i}\left(e_{1},\ldots e_{n}\right)\right)$$
and the Euclidean distance between the $gaze_{i}\left(x,y\right)$ and $T_{i}^{\left(c_{i}\right)}\left(x,y\right)$ is calculated. Then *removalPercent* of pairs with the greatest distance are removed from the dataset.

Removal of outliers is of course a broad topic and many algorithms may be used for this purpose. The RANSAC algorithm [48] is an example of a commonly used approach, which we tested during our studies. However, because the results after its usage were not significantly better than after using the approach presented above, we finally decided to use the latter as it is simpler and faster.

### 2.5. Classic Calibration Using ETCAL Library

The last step is the usage of the mapping to build a calibration model. When the training set consists of pairs of eye tracker output ($(E_{i}\left(e\right)$) and the corresponding one target ($T_{i}^{\left(c_{i}\right)}$) it is possible to use these pairs as a training set and build a calibration model that is able to calculate gaze point ($G_{i}\left(e\right)$) for every possible eye tracker output ($E_{i}\left(e\right)$).

There are many possible methods that may be used for this purpose, starting from polynomial approximation to nonlinear methods such as Support Vector Regression or Neural Networks [12]. The ETCAL library [30], which was created by the authors of the paper, implements both polynomial and SVR algorithms. The library allows choosing the specific algorithm and training its parameters. However, for simplicity, we decided to skip the training phase and use a classic quadratic polynomial function. The task was therefore to use the training set to calculate parameters of two quadratic equations that may be later used to find horizontal and vertical gaze coordinates. When there are only two eye tracker output values (like e.g., x and y coordinates of eye center) there are 12 parameters of the equation (Equation (Equation 17)):(17)$$\begin{array}{cc} g_{x}= & A_{x}e_{x}^{2}+B_{x}e_{y}^{2}+C_{x}e_{x}e_{y}+D_{x}e_{x}+E_{x}e_{y}+F_{x}\\ g_{y}= & A_{y}e_{x}^{2}+B_{y}e_{y}^{2}+C_{y}e_{x}e_{y}+D_{y}e_{x}+E_{y}e_{y}+F_{y} \end{array}$$
where $g_{x}$ and $g_{y}$ are gaze point coordinates. For more *E* values the equation becomes more complicated; however, the ETCAL library can solve it using the LM algorithm [47].

## 3. Experiments

Two datasets recorded with two types of eye trackers: the remote one, placed under a display (The Eye Tribe Tracker, The Eye Tribe, Copenhagen, Danmark) and the head-mounted one (Pupil Headset, Pupil Labs, Berlin, Germany), were prepared to verify the correctness and performance of the proposed methods (Figure 6).

For both eye trackers, we registered eye movements of participants looking at predefined stimulations consisting of static images. To preserve generality of the results there were different sets of images and different groups of participants involved in each experiment.

After collecting data, the next step was to compare the implicit calibration with the classic calibration. Our assumption was that, even when we did not expect the implicit calibration to outperform the classic one, the results would be comparable.

We compared 11 ways of producing PFTs: with the usage of expert knowledge (EXP), ground truth (GT) and nine saliency models. For the given PFTs we searched for the best mapping using both genetic and incremental algorithms and four MCFs described above: REG, DIST, DISTC and DIR. Additionally, we checked the results for various fusions of these functions and *removalPercent* values.

### 3.1. Head-Mounted Eye Tracker

The first experiment aimed at checking if the proposed implicit calibration algorithms may be used for a headset eye tracker equipped with two cameras: eye camera and scene camera. Twelve participants took part in the experiment. The participants task was to look at the stimulation displayed on the 21-inch display. They were unconstrained and could move their heads freely.

The stimulation presented on the display consisted of the following steps:Initial classic nine points calibrationReference screen 134 images—two seconds per imageReference screen 234 images—two seconds per imageReference screen 3

There were nine, equally distributed points presented on each reference screen. The task for participants was to look at the point and click it with the mouse. Every point had to be clicked twice. The results were later used to calculate errors of calibration algorithms. The whole stimulus presentation has been published and is available online [49]. The result of every session was a set of frames taken from a scene camera and eye camera. At first, every scene frame was processed to find screen corners and the corresponding eye-camera frame was processed to find eye center coordinates. As the eye camera was always in the same position in relation to the participant’s eye, the eye center coordinates could be used as eye tracker output sufficient to calculate gaze position. Therefore, eye tracker output E is denoted as $eye(x_{e},y_{e})$ for this experiment.

Eye center coordinates in the eye-camera frame were found after a pipeline of transformations: conversion to monochrome, histogram equalization, binarization, dilatation, erosion, and finally Canny edge detection. Screen corner coordinates in the scene camera frame were found after calculating the frame image integral and searching for corner-like patterns in the scene. To accelerate the algorithm, new corners were searched only in the neighborhood of the corners found in the previous frame.

In the next step, the coordinates of PFTs were recalculated into a scene coordinate system for every scene frame containing an image (see Figure 7). The last step was aggregation of frames into fixations. A sequence of frames was treated as a fixation when there were at least five subsequent eye images (about 150 ms) with eye center locations closer than five points (in eye-camera coordinates) which is equivalent to less than 0.5 deg. Such frames were aggregated into one fixation. Finally, the list of fixations (as defined in Equation (Equation 2)) was created for every trial.

The *k* frames that were recorded during the initial calibration were collected separately with information regarding the calibration point’s position cp(x,y) recalculated into scene coordinates and saved as *calPoints*:(18)$$\begin{array}{c} calPoints:\{\left(E_{1}\left(e_{x},e_{y}\right),cp_{1}\left(x,y\right)\right),\ldots ,\left(E_{k}\left(e_{x},e_{y}\right),cp_{k}\left(x,y\right)\right)\} \end{array}$$

Similarly, the *l* frames recorded during the reference screens when the participant task was to look at a specific reference point rp(x,y) were saved as *refPoints*:(19)$$\begin{array}{c} refPoints:\{\left(E_{1}\left(e_{x},e_{y}\right),rp_{1}\left(x,y\right)\right),\ldots ,\left(E_{l}\left(e_{x},e_{y}\right),rp_{l}\left(x,y\right)\right)\} \end{array}$$

The *calPoints* were later used to calculate the reference results of explicit calibration. The *refPoints* were used as the GT to calculate errors of different calibration models.

All other frames (e.g., recorded during information screens presentations) were skipped.

To use the gathered data for the implicit calibration it was necessary to define PFTs for each of the 68 images. There were ground-truth PFTs found using data taken from genuine user observations (GT), PFTs defined by an expert (EXP) and PFTs calculated by means of nine different saliency models (see Figure 4 as an example).

Incremental and genetic algorithms were used with four different MCFs. To find the best mapping the genetic algorithm was run with 1000 iterations and stopped after 200 iterations if no progress was visible. The *cutoff* value for the incremental algorithm was set to 100. Additionally, all calculations were performed with various *removalPercent*: 0, 10, 20, 30 and 40.

Every calibration model was used to calculate gaze positions for refPoints on the reference screens. The obtained values $g_{i}\left(x,y\right)$ were then compared with $rp_{i}\left(x,y\right)$—genuine gaze locations for each refPoint (Equation (Equation 19)). As the scene camera resolution was 640 × 480, the difference in scene camera pixels was recalculated to a percentage of the scene resolution using the formula:(20)$$\begin{array}{c} Error(x)=100*|rp(x)-g(x)|/640\\ Error(y)=100*|rp(y)-g(y)|/480 \end{array}$$

Errors for all refPoints where then averaged.

Reporting error in the scene-size percentage is in our opinion the most intuitive way when Human-Computer interfaces are taken into account (and this is the main application of the presented solution). However, recalculation of those values to degrees is possible after estimating the scene camera viewing angle [50]. Our camera viewing angle, as reported by the manufacturer, was 60 degrees, but our measurements revealed it to be 70 degrees horizontally and 53 degrees vertically. Therefore, the errors presented in the Results section may be recalculated to degrees by multiplying the horizontal percent error by 0.7 and the vertical error by 0.53.

### 3.2. Remote Eye Tracker

The Eye Tribe eye tracker registering eye movements with a frequency of 60 Hz was used in the second experiment. As this eye tracker needs to be calibrated to start working, the calibration procedure was initially started with one person who did not take part in the later experiments. Therefore, the signal obtained from the eye tracker could be treated as an uncalibrated signal in the further experiments.

The stimulation consisted of the following steps:Initial 9-point calibration6 images observations (3 s each)Reference screen—user clicks 9 points looking at them6 images observations (3 s each)Final 9-point calibration

Twenty-nine participants took part in the experiment. The eye movement signal recorded during the reference screen presentation for each of them was used to check the implicit calibration algorithm. The initial and final calibrations were done to enable comparison of our method to the standard calibration. All errors were calculated as the difference between the calibrated gaze locations and locations computed using data from the reference screen. Due to different screen resolutions for each trial and to maintain consistency with the previous experiment, the error is reported as a percentage of screen dimensions independently for X and Y axes. These values may be converted to degrees only when the distance between the screen and the observer is known, which enables calculation of the screen size in degrees [50]. Unfortunately, we did not collect such information during every trial. However, if we assume that this distance was approximately 60 cm, it may be derived that the screen covered about 40 degrees horizontally and 32 degrees vertically in the observer’s field of vision. Therefore, a rough recalculation to degrees requires multiplying the horizontal errors by 0.4 and vertical errors by 0.32.

To use the gathered data for implicit calibration it was necessary to define PFTs for each of the 12 images. As in the previously described experiment, the PFTs were identified by means of data taken from: genuine users’ observations (GT), PFTs defined by an expert (EXP) and PFTs calculated using nine different saliency models. Incremental and genetic algorithms were used with four different MCFs and all calculations were performed with various *removalPercent*: 0, 10, 20, 30 and 40.

### 3.3. Movie Experiment

The Eye Tribe eye tracker registering eye movements with a frequency of 60 Hz was also used in the third experiment. This time 20 participants were watching a movie. The movie lasted for 66 s and presented part of an animation movie ‘Bolek i Lolek’ (see Figure 8). Similarly to the previous experiment, the simulation consisted of several steps:Initial 9-point calibrationMovie watchingReference screen—user clicks 9 points looking at them

The whole video sequence is available online [51].

The first processing step was the extraction of frames from the movie. Five frames were extracted from each second of the video giving the overall number of 328 frames. The next step was the calculation of saliency maps for each frame. All nine saliency models and the GT fixation map were used. Then the targets were extracted for each frame. The rest of the calculations was the same as for the previous experiments.

## 4. Results

This research aimed to check if the described implicit calibration algorithms may give results usable in practical applications. Therefore, the data gathered during all three experiments was used to build various calibration models and calculate the calibration errors.

### 4.1. Head-Mounted Eye Tracker

The first step was checking which value of *removalPercent* gives the lowest error for GT saliency. The results are presented in Table 1. The differences were not significant and 20% was chosen for subsequent tests.

The next task was checking the results of a classic calibration. The calPoints (Equation (Equation 18)) were used as input to the LM algorithm calculating calibration function parameters. The errors are presented as CALIB in Table 2.

To check the baseline calibration—without the PFTs defined—we decided to additionally build a calibration model that uses the ‘center location paradigm’ stating that most of the time observers look at the center of a screen. Therefore, for each fixation we created an artificial target in the middle of a screen and we built the mapping that takes into account only these targets. The errors of such a calibration are denoted as CENTER in Table 2.

Finally, the calibration models were built using various MCFs and both optimization algorithms. The errors for GT and EXP saliencies are presented in Table 2.

It is important to note that, because in this case the image from the eye camera is flipped horizontally in relation to the scene camera image, it was necessary to use Equation (Equation 12) instead of (Equation 11) in $MCF_{DIR}$ calculation.

As the MCFs using DIR occurred to be significantly better, only $MCF_{DIR}$ calculated using genetic algorithm was used in the next step, which involved comparison of different saliency models (Table 3).

The GT and EXP models are not surprisingly the best ones; however, RARE2012, SWD, GBVS, and BMS models give errors that are almost the same. To check the significance of these differences, we used the *t*-Student test. The test was calculated for every pair of saliency models separately for both *x* and *y* axes, and the difference between them was defined as significant only if the difference for one of the axes occurred to be significant (with *p*-value < 0.001) (see Table 4). It may be noticed that three groups of models may be distinguished—(1) RARE2012 and GBVS which are not significantly worse than GT and EXP; (2) SWD, BMS, SR-PQFT and IttiKoch which are significantly worse than GT and EXP but still not significantly worse than the first group; and (3) LDS, FES and CovSal which are clearly the worst ones and their usage in this application is questionable. The most probable reason it the fact that there were very few targets found using these models. However, there is no apparent correlation between the number of targets and the error—the best GT defines the highest number of targets but, at the same time, the number of targets for EXP model is one of the lowest.

The next step was checking various fusions of MCFs. As $MCF_{DIR}$ appeared to be the best option, all fusions included this type of MCF complemented by one of the other MCFs with various weights from 0.2 to 0.4. The results are presented in Table 5, only for fusions that resulted in an average error less than $MCF_{DIR(g)}$. It occurred that only fusions using the genetic algorithm were able to improve the result. Additionally, the fusions of $MCF_{DIR}$ with $MCF_{DISTC}$ occurred to give the best results. It seems that—despite not good results as a single criterion—the $MCF_{DISTC}$ function is the most valuable addition to $MCF_{DIR}$ criterion when a fusion is taken into account.

### 4.2. Remote Eye Tracker

Similarly to the previous experiment, we checked which value of the *removalPercent* gives the best results when taking into account targets defined by genuine gazes (GT).

The best results were achieved for the *removalPercent* equal to 20%, so all subsequent results are reported taking into account only tests with such value (Table 6).

The baseline for evaluating the implicit calibration results were errors for two classic calibrations performed at the beginning and end of the trial. The error values for the classic calibrations are presented as CALIB1 and CALIB2 in Table 7.

The center location paradigm was used as another baseline to evaluate implicit calibration errors. The mapping that takes into account only targets in the middle of the screen was denoted as CENTER in Table 7.

Table 7 presents error values of different MCFs in relation to the baselines described above. The errors were calculated for GT and EXP defined PFTs.

It occurred that MCFs using the DIR algorithm are significantly better than other methods. When we chose the best MCF (DIR G) and 20% removal percent it was possible to compare different saliency maps (Table 8).

We may observe that once again EXP and GT models give the best results. This time only SR-PQFT and BMS models are comparable with them. To check the significance of these differences, we performed the *t*-Student test for every pair and treated the difference as significant only if the test gave a *p*-value < 0.001 for either horizontal or vertical axis. The results are presented in Table 9. When comparing these results with the results for the head-mounted eye tracker (Table 4), we may observe that: (1) LDS, CovSal and FES models are consistently the worst ones; (2) RARE, GBVS, and BMS models are still among the best ones (however the difference between them and the EXP model this time is significant); (3) The results of SWD model are worse and the results of SP-PQFT are better than previously. The numbers of targets are similar for each model with an exception for SWD (the number is higher) and CovSal and FES (the numbers are lower). However, there is no strong correlation between errors and the number of targets.

As in the previous experiment with the head-mounted eye tracker, the next step was to search for the best fusions of different criteria. As $MCF_{DIR}$ appeared to be the best option, all fusions used this type of MCF and additionally added one of the other MCFs with various weights from 0.2 to 0.4. The best results are presented in Table 10.

### 4.3. Movie Experiment

The results for the third experiment are presented as for the previous ones. The comparison of different saliency models is presented in Table 11. The results are worse than for the experiment with static images; however, they are comparable for automatically generated saliency maps. Interestingly, the model based on real gazes was not able to perform satisfactorily. It must be remembered that using classic saliency models when the stimulus is an animated movie carries certain limitations:The models do not take into account the contents of the previous frames, while such information may be important to estimate points of interest correctly.All models used in this research were prepared for real-world images (for instance photographs of nature or people), so their performance during the animated movie with artificial and sharp contours may be far from optimal.

Probably the type of images (drawings instead of natural images) resulted also in the lower number of targets found for each image. Nevertheless, the results show that the implicit calibration based on PFTs is usable even in such a case.

### 4.4. Time Analysis

The ultimate goal of this project is to prepare an algorithm that will be able to work online, during the usage of an eye tracker. Therefore, time analysis is an essential part of the work.

Computation times for the saliency maps are demonstrated in Table 12. These are average values of computation time per frame for the test movie (see Figure 8). The testing machine was Lenovo ThinkStation S20 with Xeon W3550 at 3.06 GHz CPU, 12 GB RAM.

The results are not impressive as only two of the models offered more than five frames per second. However, it should be noted that all but one of the implementations were executed using the MATLAB environment (Matlab 2016b working on Windows 10). It may be supposed that the results would be much better if a compiled code has been used. Moreover, as shown in the movie experiment, even five frames per second is quite enough for movie frames analysis.

The next time-consuming step was mapping calculation. There were two optimization algorithms used: genetic and incremental ones and four MCF. Table 13 presents the time required to calculate mapping when all the data from dataset 2 (remote eye tracker with static images) was used. It is visible that only $MCF_{DISTC}$ and $MCF_{DIR}$ algorithms offer efficiency suitable for the online calibration.

The results presented above were calculated for all fixations and all targets available. Therefore, an interesting question was how many fixations are necessary to obtain reliable results. The $MCF_{DIR}$ and incremental algorithm were used to assess how the error changes when the number of fixations increases. The results are presented in Figure 9 for the remote eye tracker and in Figure 10 for the head-mounted eye tracker.

It is visible that after analysis of approximately 50 fixations (20–30 s depending on the trial) the error stabilizes.

## 5. Discussion

This paper presents an implicit calibration algorithm that uses the idea of PFTs. Using the targets instead of the whole saliency maps as was typically done in previous research [17,22,24] has several important advantages:Data processing is easier and faster, which makes a ‘real-time’ calibration feasible.It is more universal as it may be used with any stimulus: static images, movies—but also with computer interfaces (e.g., with buttons or labels as targets) or games (e.g., with avatars as targets as was proposed in [6]).Creating a short list of potential targets could be faster than the creation of a full saliency map. For example a face extractor using fast Haar cascades may be used to define each face as a potential target.Definition of targets may be easily done manually, by a human expert (which was checked experimentally) while the creation of a saliency map requires special algorithms.

The results of three experiments described in the previous section show that the implicit calibration using PFTs may be used instead of classic calibration. Error levels are higher than for the classic calibration; however, these levels are comparable for some MCFs.

It is challenging to reliably compare the achieved results to results of other implicit calibration algorithms. On one hand implementation of their algorithms is often not possible due to an incomplete description, on the other hand, the datasets for which the experiments were performed are typically not available. The comparison of bare results given in degrees when different data registered using different eye trackers and with different preprocessing algorithms were used is in our opinion not reliable. Nevertheless, it may be concluded that our method, with errors less than 2 degrees for static images and less than 4 degrees for a movie is comparable with other state-of-the-art methods. For example Sugano et al. [28] reports errors over 5 degree, Alnajar et al. [24] about 4 degrees, Chen et al. [17] less than 3 degrees.

Unsurprisingly, the best results were achieved for GT and expert-based targets. However, for some fully automatic saliency models, the results were not much worse than for the GT (and even better for the movie experiment). Specifically, the results for the RARE2012, GBVS and BMS models occurred to be similarly acceptable for all three experiments, while the results for the LDS, FES and CovSal models occurred to be unacceptable. Probably these outcomes may be improved, when more sophisticated algorithms for targets extraction will be used—which may become an interesting direction for further research.

Of course, the quality of the PFT-based calibration, as for every implicit calibration, depends heavily on the type of stimulus being presented. If the probable targets of stimuli are always near the center of a screen (which is e.g., typical for movies), the quality of the calibration model near the edges of the scene will decrease. This is the main limitation regarding the presented solution. However, the same problem applies to every implicit calibration algorithm and if all probable targets are indeed gathered near the center of the screen, then the calibration model should also focus on the center and its accuracy near the edges may be not so important. It may be observed in the video created for one participant, comparing ground-truth gazes (red circle) with the gazes calculated with the usage of RARE2012 saliency model (cyan circles) [51]. It occurs that the calculated gaze is quite accurate near the center of the screen and the accuracy drops when the gaze goes towards the screen edges.

Another limitation is that errors obtained for implicit calibration using saliency models are so far significantly higher than for the classic calibration. However, there is still a lot of room for improvement. As for the future work we plan to investigate how to improve the PFTs generation algorithm that will find more robust PFTs by extracting it from saliency maps. We also plan to test new fusions of already introduced MFCs and search for new MFCs. Finally, we would like to test the method using new datasets with different stimuli such as movies or classic user interfaces.

To maintain the repeatability of the research the code and the datasets were published online at www.kasprowski.pl/etcal (Appendix A). 

## Figures and Tables

**Figure 1 sensors-19-00216-f001:**
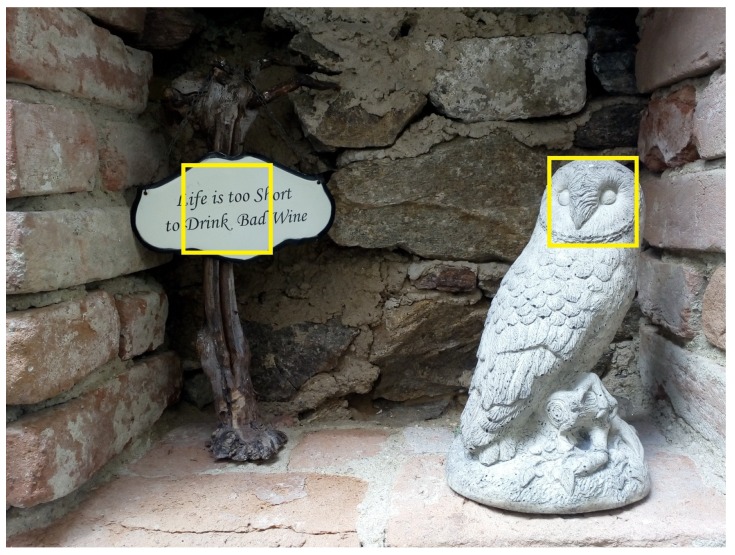
The image with two probable fixation targets (PFTs). It may be assumed that while looking at the image, most of the time people will fixate on one of the regions inside the yellow squares.

**Figure 2 sensors-19-00216-f002:**
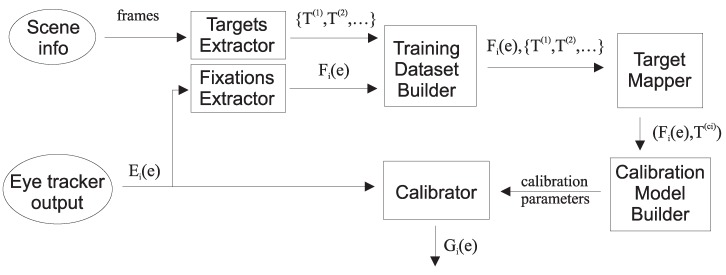
Data flow schema of the implicit calibration algorithm taking into account PFT.

**Figure 3 sensors-19-00216-f003:**
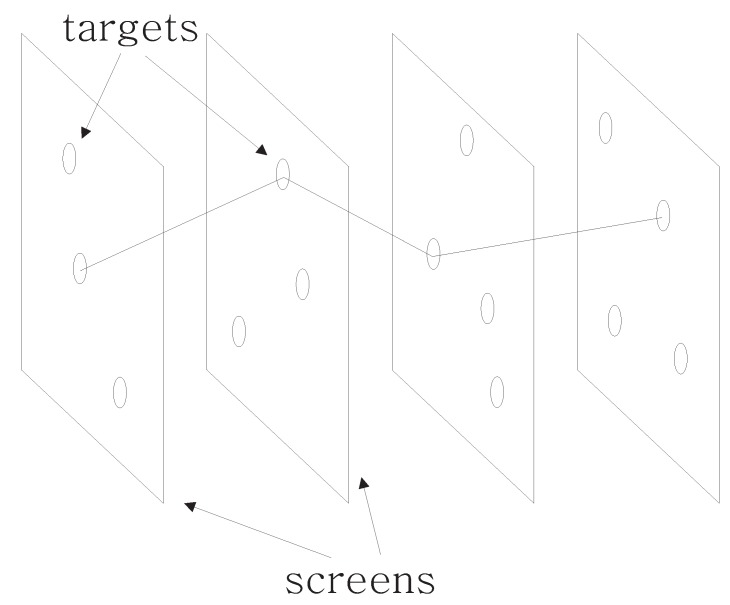
Fixations with targets. The figure shows subsequent scenes/screens with several targets (PFTs) on each. Horizontal axis represents time. The path visualizes one possible mapping.

**Figure 4 sensors-19-00216-f004:**
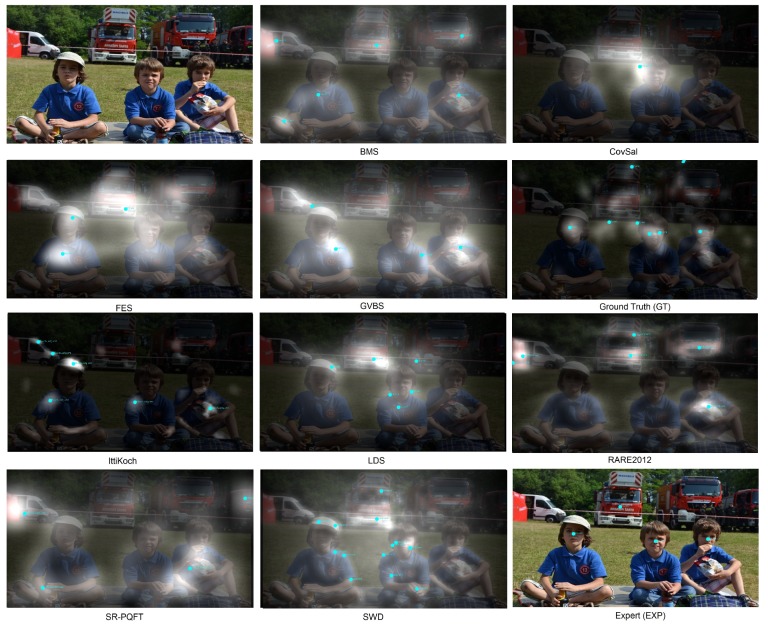
Exemplary image with corresponding saliency maps obtained with different methods. The targets (PFTs) are presented as cyan dots with labels on each image.

**Figure 5 sensors-19-00216-f005:**
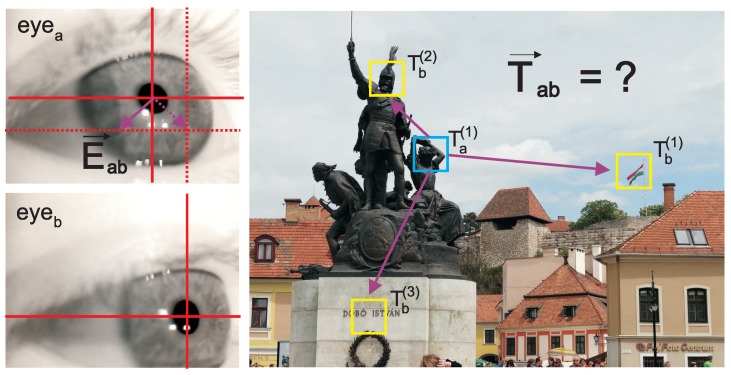
The participant was looking at $T_{a}^{\left(1\right)}$ during fixation *a* and the eye center was in the point shown in $eye_{a}$ image. Having fixation *b* with the eye center in the point shown in $eye_{b}$ image it may be assumed that the correct gaze target should be $T_{b}^{\left(3\right)}$, because it results in the lowest angle between $\vec{E_{ab}}$ flipped horizontally (Equation (Equation 12)) and $\vec{T_{ab}}$ (Equation (Equation 13)).

**Figure 6 sensors-19-00216-f006:**
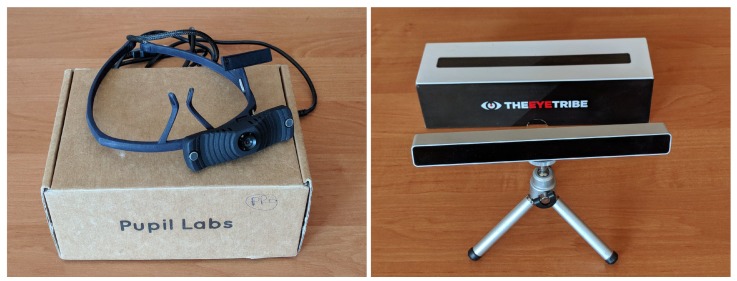
Eye trackers used during the experimental part of the research. Left: Pupil Labs headset eye tracker, right: The Eye Tribe remote eye tracker to be mounted below a screen.

**Figure 7 sensors-19-00216-f007:**
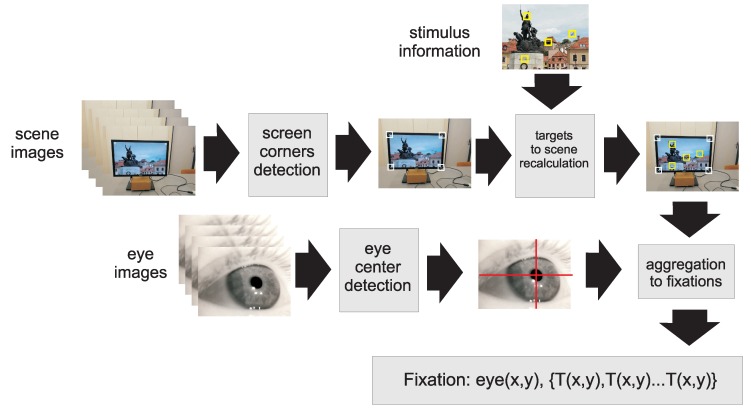
A pipeline of image processing converting scene and eye frames to a list of fixations with eye and targets coordinates.

**Figure 8 sensors-19-00216-f008:**
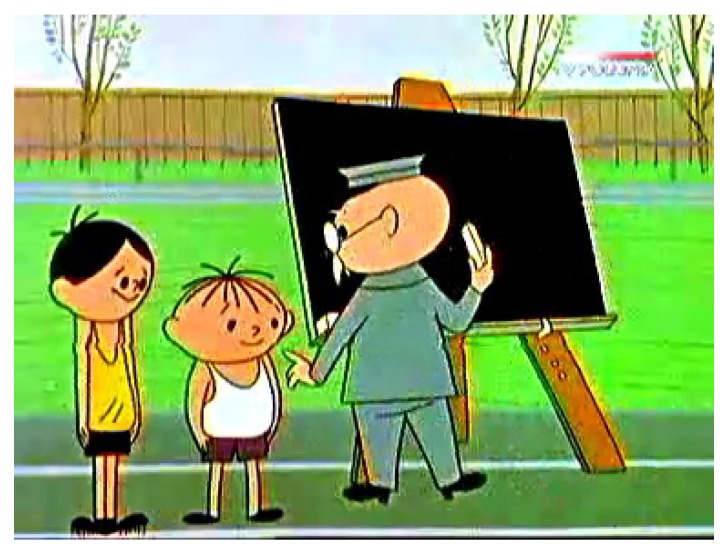
A frame from the movie presented during the third experiment.

**Figure 9 sensors-19-00216-f009:**
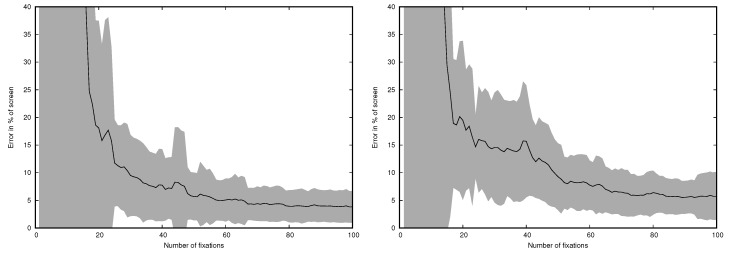
Error for the remote eye tracker with $MCF_{DIR}$ and incremental algorithm depending on the number of fixations. Left: horizontal error, right: vertical error. The gray area represents standard deviation.

**Figure 10 sensors-19-00216-f010:**
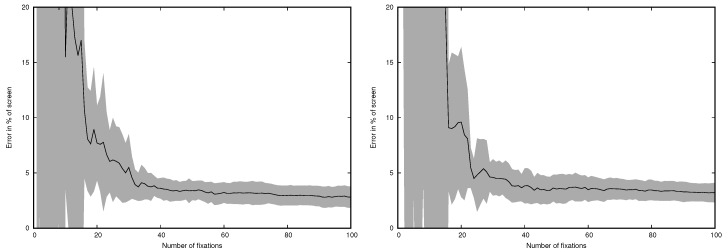
Error for the headset eye tracker and $MCF_{DIR}$ incremental algorithm depending on the number of fixations. Left: horizontal error, right: vertical error. Gray area represents standard deviation.

**Table 1 sensors-19-00216-t001:** Average errors for various *removalPercent* values.

Removal Percent	Error X (%)	Error Y (%)	Avg Error (%)
0	8.52	7.17	7.84
10	8.11	7.06	7.59
20	8.06	7.03	7.55
30	8.12	7.08	7.60
40	8.13	7.23	7.68

**Table 2 sensors-19-00216-t002:** Average errors for various MCFs with GT and expert (EXP) saliency models. CALIB stands for classic explicit calibration and CENTER for center target calibration. Algorithm G—genetic optimization, I—incremental optimization.

MCF	Algorithm	Saliency Model	Error X (%)	Error Y (%)	Avg Error (%)
CALIB			2.49	2.02	2.26
DIR	G	GT	2.36	2.82	2.59
DIR	I	GT	2.40	3.15	2.77
DIR	I	EXP	2.65	2.97	2.81
DIR	G	EXP	2.71	2.94	2.83
REG	G	EXP	5.76	6.92	6.34
DIST	I	GT	7.72	5.63	6.68
REG	I	EXP	8.09	7.19	7.64
DIST	I	EXP	9.69	5.74	7.71
DC	G	EXP	8.76	7.33	8.05
DIST	G	EXP	9.54	6.94	8.24
DISTC	I	EXP	9.00	7.55	8.28
DISTC	I	GT	8.28	8.52	8.40
DC	G	GT	9.02	8.58	8.80
DIST	G	GT	10.64	8.80	9.72
REG	G	GT	11.53	9.06	10.30
REG	I	GT	12.49	9.72	11.11
CENTER			12.41	10.03	11.22

**Table 3 sensors-19-00216-t003:** Errors for various saliency models when $MCF_{DIR}$ is used. The last column shows the average number of targets per image found for the given saliency model.

Saliency Model	Error X (%)	Error Y (%)	Avg Error (%)	Number of Targets
GT	2.36	2.82	2.59	10.38
EXP	2.71	2.94	2.83	3.47
RARE2012	3.18	3.36	3.27	4.79
SWD	3.48	3.68	3.58	9.25
GBVS	3.05	4.21	3.63	3.47
BMS	4.03	3.52	3.78	5.12
SR-PQFT	4.99	3.64	4.31	4.12
IttiKoch	4.12	4.98	4.55	4.24
LDS	7.95	7.10	7.52	1.78
FES	8.07	7.91	7.99	1.74
CovSal	10.63	8.91	9.77	1.49

**Table 4 sensors-19-00216-t004:** Significance of differences of calibration errors between various saliency models taking $MCF_{DIR}$ and *removalPercent* = 20. The asterisk (∗) denotes that according to the *t*-Student test for either *x* or *y* axis the difference was significant (*p* < 0.001).

	GT	EXP	RARE2012	GBVS	SWD	BMS	SR-PQFT	IttiKoch	LDS	FES	CovSal
GT					∗	∗	∗	∗	∗	∗	∗
EXP							∗	∗	∗	∗	∗
RARE2012									∗	∗	∗
GBVS									∗	∗	∗
SWD	∗								∗	∗	∗
BMS	∗								∗	∗	∗
SR-PQFT	∗	∗							∗	∗	∗
IttiKoch	∗	∗							∗	∗	∗
LDS	∗	∗	∗	∗	∗	∗	∗	∗			∗
FES	∗	∗	∗	∗	∗	∗	∗	∗			
CovSal	∗	∗	∗	∗	∗	∗	∗	∗	∗		

**Table 5 sensors-19-00216-t005:** Average errors for various MCF fusions calculated for GT saliency and *removalPercent* = 20%. Only fusions that gave results better than the baseline $MCF_{DIR(g)}$ are listed.

MCF	Algorithm	Error X (%)	Error Y (%)	Avg Error (%)
DIR+0.3×DISTC	G	2.24	2.62	2.43
DIR+0.2×REG	G	2.17	2.69	2.43
DIR+0.2×DISTC	G	2.24	2.69	2.47
DIR+0.4×DISTC	G	2.29	2.66	2.47
DIR+DISTC	G	2.29	2.71	2.50
DIR+0.4×REG	G	2.34	2.71	2.53

**Table 6 sensors-19-00216-t006:** Average errors for various *removalPercent* values for GT saliency.

Removal Percent	Error X (%)	Error Y (%)	Avg Error (%)
0	14.81	12.51	13.66
10	14.16	11.80	12.98
20	13.97	11.35	12.66
30	14.05	11.43	12.74
40	14.42	11.48	12.95

**Table 7 sensors-19-00216-t007:** Average errors for GT PFTs and various mapping comparison functions (MCF).

MCF Type	Algorithm	Saliency	Error X (%)	Error Y (%)	Avg Error (%)
CAL2			3.04	4.21	3.63
CAL1			3.04	4.59	3.82
DIR	G	EXP	3.44	5.37	4.41
DIR	I	EXP	3.44	5.77	4.60
DIR	G	GT	4.85	6.50	5.68
DIR	I	GT	5.35	6.41	5.88
REG	G	EXP	8.44	7.70	8.07
REG	G	GT	10.15	11.41	10.78
REG	I	EXP	12.65	9.51	11.08
DIST	I	GT	13.64	12.16	12.90
DIST	I	EXP	16.42	10.38	13.40
DISTC	G	EXP	16.91	13.23	15.07
DIST	G	GT	19.88	11.53	15.70
DISTC	I	EXP	17.94	14.08	16.01
DISTC	G	GT	19.48	12.69	16.08
DISTC	I	GT	18.28	14.87	16.58
REG	I	GT	20.33	15.33	17.83
DIST	G	EXP	22.27	15.35	18.81
CENTER			21.37	18.41	19.89

**Table 8 sensors-19-00216-t008:** Average errors for various saliency models taking $MCF_{DIR}$ and *removalPercent* = 20.

MCF Type	Error X (%)	Error Y (%)	Average Error	Number of Targets
EXP	3.44	5.37	4.41	3.07
GT	4.85	6.50	5.68	4.62
SR-PQFT	5.89	7.78	6.84	3.71
BMS	4.87	9.53	7.20	4.79
GBVS	5.94	10.19	8.07	4.43
RARE2012	4.77	12.29	8.53	5.43
IttiKoch	9.18	8.21	8.70	5.36
SWD	8.90	8.51	8.71	9.71
LDS	12.91	11.46	12.18	2.29
CovSal	19.18	18.37	18.78	1.36
FES	20.83	17.20	19.02	1.36

**Table 9 sensors-19-00216-t009:** Significance of differences of calibration errors between various saliency models taking $MCF_{DIR}$ and *removalPercent* = 20. The asterisk (∗) denotes that according to the *t*-Student test for either *x* or *y* axis the difference was significant (*p* < 0.001).

	EXP	GT	SP-PQFT	BMS	GBVS	RARE2012	IttiKoch	SWD	LDS	CovSal	FES
EXP				∗	∗	∗	∗	∗	∗	∗	∗
GT						∗	∗	∗	∗	∗	∗
SP-PQFT								∗	∗	∗	∗
BMS	∗							∗	∗	∗	∗
GBVS	∗								∗	∗	∗
RARE2012	∗	∗					∗	∗	∗	∗	∗
IttiKoch	∗	∗				∗				∗	∗
SWD	∗	∗	∗	∗		∗			∗	∗	∗
LDS	∗	∗	∗	∗	∗	∗		∗		∗	∗
CovSal	∗	∗	∗	∗	∗	∗	∗	∗	∗		
FES	∗	∗	∗	∗	∗	∗	∗	∗	∗		

**Table 10 sensors-19-00216-t010:** Average errors for various fusions of MCFs, saliency EXP and *removalPercent* = 20. Only the best achieved fusions are presented.

MCF Type	Algorithm	Number of Trials	Error X (%)	Error Y (%)	Average Error
DIR+0.2×DIST	G	29	3.00	4.83	3.91
DIR+0.2×REG	G	29	2.97	4.86	3.91
DIR+0.4×REG	G	29	2.93	5.03	3.98
DIR+0.2×DISTC	G	29	3.07	5.00	4.03
DIR+0.3×DISTC	G	29	2.93	5.14	4.03
DIR+0.4×DISTC	G	29	2.79	5.28	4.03
DIR+0.3×REG	G	29	2.93	5.14	4.03
DIR+0.4×DIST	G	29	3.00	5.14	4.07

**Table 11 sensors-19-00216-t011:** Average errors for various saliency models taking $MCF_{DIR}$ and *removalPercent* = 20.

MCF Type	Error X (%)	Error Y (%)	Average Error	Number of Targets
CAL	3.44	3.90	3.67	3.35
RARE2012	7.38	11.51	9.45	4.35
IttiKoch	10.22	9.48	9.85	2.46
GT	12.92	9.76	11.34	1.38
GBVS	9.00	15.16	12.08	2.14
SR-PQFT	12.06	14.32	13.19	3.62
BMS	13.27	14.13	13.70	6.29
FES	14.31	13.24	13.77	2.13
SWD	17.59	15.75	16.67	1.74
LDS	19.71	15.80	17.75	1.38
CovSal	19.82	17.35	18.58	1.62

**Table 12 sensors-19-00216-t012:** Average execution time for saliency maps computation per frame of movie in the third experiment.

Saliency Map	Implementation	Average Execution Time (s)
FES	MATLAB	0.170
BMS	exe (binary)	0.196
SWD	MATLAB	0.242
Itti-Koch	MATLAB	0.328
SR-PQFT	MATLAB	0.388
LDS	MATLAB	0.511
RARE2012	MATLAB	0.746
GBVS	MATLAB	0.921
CovSal	MATLAB	21.4

**Table 13 sensors-19-00216-t013:** Average times of execution for different MCFs for remote eye tracker.

MCF Type	Incremental Time (s)	Genetic Time (s)
DISTC	0.10	2.32
DIR	3.78	9.52
REG	26.86	62.24
DIST	127.64	492.41

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
