# Peer review of "Implicit Calibration Using Probable Fixation Targets"

_sensors, 2019, doi:10.3390/s19010216_

Round 1

Reviewer 1 Report

The authors present and evaluate an implicit method for an eye tracker calibration. The method utilizes a saliency map of presented content to build a set of probable fixation targets. Optimization then finds the best correspondences between eye tracker outputs and the targets and a simple polynomial model is used to find the best mapping between them. The authors compare different version of the algorithm in two user studies.

The motivation of the method is clear and the results show that the method can be practical in certain scenarios. However, I am little bit unsure how to judge the paper. The authors acknowledge that it is an extension of their ETRA paper [24] and list the new contributions separately. However, as far as I know this is a new separate paper and the previous work is already considered published. This means that I only review the novel contributions. Unfortunately, I find them too limited. It is hard to judge the extent of the new work because the paper repeats too much content from the previous publication. I believe it is not a plagiarism as it is openly admitted but it is unnecessary. I recommend authors to rewrite the paper so that it focuses on presentation of new results and uses citation and conciseness when referring to techniques already presented in the previous publications.

As far as I can tell, the main contributions are two new mapping comparison functions which are, however, performing worse than the original "DIR", then the second user study with a remote eye tracker and finally evaluation of saliency prediction methods as determinants of the probable fixations. I believe the last contribution is the most interesting one and probably should get more space. I am not sure if the first study is the same one as in [24] because the description seem similar including participants and procedure but the results differ slightly even for the explicit calibration. It should be clarified. If it is the same study then I believe citing it is enough and focus can be put on the new one.

Authors mention several other implicit calibration methods that use saliency in a continuous way ([11,16,18]). Adding a comparison to these techniques would strengthen the submission. It is not evident why the proposed method is faster then the continuous saliency methods if the fixation targets are built from already computed saliency maps (L524). A comparison would answer such questions.

There is no discussion of differences between outcomes of both studies in this paper. Although the mapping function "DIR" performs consistently well, the order of the saliency methods is completely different in each of the two studies. Does it mean the differences are statistically insignificant or is there a reason for different gaze pattern of subjects being tracked by one or the other device? It does not seem intuitive to me since the presented content and the experiment procedure should be the same. If both studies are a new contribution then I would describe them in a more joint way and present their results side-by-side for comparison.

The method seems to be limited to static images as the construction and mapping of fixation targets require extended viewing of a single content. Authors propose testing the method on movies (L548). How can this be done? An extension for dynamic content would definitely be a solid technical contribution.

* References *

References are adequate and the discussion of previous work is a strong part of the paper.

* Presentation *

The presentation and language style seem adequate to me. My main complain is about the extent of text that is dedicated to the description of the techniques from previous work such as the two optimization algorithms. It disolves the novelty of the paper.

* Reproducibility *

The method seems reproducible. It should be clarified how it is selected which (local?) minima to accept as potential targets in the watershed algorithm (L212). I assume that for a noisy saliency map there can be quite many. Please also clarify the viewing distance in the experiment with the remote tracker.

* Minor issues * 

- Figure 3: It is very hard to associate the saliency maps and the image content. It would be better to use an overlay.

- Table 3 is not referenced and probably also not discussed. Discussion of the saliency method performance is completely missing and I believe it is an important contribution.

* Summary *

The paper is well written and apart of its length it is easy to read. The technique seems useful and it is well motivated. If all the content was novel, I would recommend a minor revision for improving the study presentation and extending the discussion of saliency methods performance. However, I believe that novelty is significantly limited by the previous work. The paper should be rewritten to reflect it and focus on the novel content. My feeling is that there is unfortunately not enough novelty at the moment. This could be improved by adding a comparison to previous work, evaluating the continuous version of the calibration process in a more realistic usage scenario or extending the method to dynamic content such as movies. This is why I recommend to reconsider the paper after a major revision.

Author Response

We would like to thank both reviewers for their valuable comments. We corrected the paper according to their remarks, specifically:

·        We added a description of another experiment that calibrates users based on movie watching data (Sections 3.3 and 4.3)

·        We added a more comprehensive discussion about saliency model results – including significance analysis (in sections 4.1 and 4.2 and in Summary)

·        We corrected the structure of the paper by moving part of the Results section to a new Experiment section placed between Methods and Results sections.

·        We added some more introductory information about current technologies and calibration methods.

Additionally, we did our best to correct any language mistakes and the paper was checked by a native speaker.

All new text (apart from language corrections) is denoted with the blue color in the current text.

Specific comments/answers to reviews:

Reviewer 1:

It is hard to judge the extent of the new work because the paper repeats too much content from the previous publication.

It is true, that the paper repeats some content of two conference publications. However, the conference publications had strict page limits and not every issue could be explained in detail. Our aim was to summarize the whole idea of implicit calibration with the use of probable fixation targets in one publication in the way that will be readable and reproducible.

I recommend authors to rewrite the paper so that it focuses on presentation of new results and uses citation and conciseness when referring to techniques already presented in the previous publications.

Our aim was to summarize the work presented in the previous publications in the way that will be more readable and add some more implementation details that make it reproducible – that is why we decided to repeat some parts of the previous conference publications. However, according to this remark, we added more new material – a more extensive discussion about the results achieved for various saliency models and a completely new dataset created during movie watching.

I believe that the evaluation of saliency prediction methods as determinants of the probable fixations is the most interesting one and probably should get more space.

According to this remark we added more extensive discussion about the results achieved for various saliency models and  significance analysis.

I am not sure if the first study is the same one as in [24] because the description seem similar including participants and procedure but the results differ slightly even for the explicit calibration

Yes, the same dataset from [24] was used in this work. The results presented in this paper are reported in screen per cent to be consistent with two other datasets and the results in [24] were reported in degrees. We decided that screen per cent is more readable result for a person not experienced in eye tracking. However, in the corrected version of the paper, we added information about the relation of both values (lines: 449-455, 474-480). As for small differences in the results it should be emphasized that the whole code was rewritten to make it compatible with the ETCAL library and all calculations were redone.

Authors mention several other implicit calibration methods that use saliency in a continuous way ([11,16,18]). Adding a comparison to these techniques would strengthen the submission.

We added a paragraph with a short comparison with other methods in Summary (lines: 628-636).

It is not evident why the proposed method is faster then the continuous saliency methods if the fixation targets are built from already computed saliency maps (L524). A comparison would answer such questions.

Our reasoning here was that targets don’t always have to be created from saliency maps. For instance a simple and fast face extractor (e.g. using Haar cascades) may be used to find each face as a probable target. We added information why we think that it may be in some circumstances faster to find PFTs than to build a saliency map (lines 620-622).

Although the mapping function "DIR" performs consistently well, the order of the saliency methods is completely different in each of the two studies. Does it mean the differences are statistically insignificant or is there a reason for different gaze pattern of subjects being tracked by one or the other device?

To address this issue we added more detailed comparison of the errors of saliency models in both experiments and information about a significance of differences between them (lines 525-536, 560-570).

The method seems to be limited to static images as the construction and mapping of fixation targets require extended viewing of a single content. Authors propose testing the method on movies (L548). How can this be done? An extension for dynamic content would definitely be a solid technical contribution.

According to this remark we added information about the third experiment that uses movie stimulus. In fact, we had this eye movement data ready already registered and would like to thank the reviewer for this valuable comment and encouragement to perform the calculations. We hope it proved that the same method may be used to dynamic content (sections 3.3 and 4.3).

It should be clarified how it is selected which (local?) minima to accept as potential targets in the watershed algorithm (L212). I assume that for a noisy saliency map there can be quite many.

The paragraph describing target extraction was indeed unclear. We added more complete information how our algorithm works (lines 222-232).

Please also clarify the viewing distance in the experiment with the remote tracker.

Unfortunately, we did not measure the exact distance for every trial. But it was possible to estimate it roughly (lines: 449-455, 474-480).

- Figure 3: It is very hard to associate the saliency maps and the image content. It would be better to use an overlay.

The Figure was corrected.

- Table 3 is not referenced and probably also not discussed. Discussion of the saliency method performance is completely missing and I believe it is an important contribution.

We added a more complete discussion about saliency models’ results (lines 525-536, 560-570). We also added information about execution times for various saliency models (lines 590-599).

Reviewer 2 Report

The paper proposes an interesting approach to a sensitive problem, which is the calibration of eye tracking devices. The experiment seems to be conducted in the proper way, and the results support the thesis from the authors. Even if the approach of implicit calibration is not new, the approach is interesting with promising results.

Nevertheless, there are a number of major issues that need to be resolved before the paper is publishable.

MAIN ISSUES:

- IN GENERAL: language is not too bad, but it is still quite far from English. Please, have the paper checked by a native speaker.

- One big risk of your method is to incur in overfitting of the subject's data. Just like what is used in neural networks problems, it would be interesting to do a random division between the TRAINING gaze data (maybe 80% of the total or so) and the TEST gaze data (the remaining 20%). In this way you can compute the calibration error on the test data set only, and have a better evaluation of the result of the calibration. Otherwise, you could the same procedure using some images as TRAINING set an the remaining as TEST set.

- INTRODUCTION and RESULTS: eye tracking is becoming very popular in the last years, it would help the clarity of the paper to provide a short introduction about the current technologies and frame them in the literature. I suggest these citations:

Dalmaijer, Edwin. "Is the low-cost EyeTribe eye tracker any good for research?." PeerJ PrePrints (2014).

Funke, Gregory, et al. "Which Eye Tracker Is Right for Your Research? Performance Evaluation of Several Cost Variant Eye Trackers." Proceedings of the Human Factors and Ergonomics Society Annual Meeting. Vol. 60. No. 1. Sage CA: Los Angeles, CA: SAGE Publications, 2016.

Mannaru, Pujitha, et al. "Performance Evaluation of the Gazepoint GP3 Eye Tracking Device Based on Pupil Dilation." International Conference on Augmented Cognition. Springer, Cham, 2017.

Gibaldi, Agostino, et al. "Evaluation of the Tobii EyeX Eye tracking controller and Matlab toolkit for research." Behavior research methods 49.3 (2017): 923-946.

Kar, A., & Corcoran, P. (2018). Performance Evaluation Strategies for Eye Gaze Estimation Systems with Quantitative Metrics and Visualizations. Sensors, 18(9), 3151.

Such an introduction it would allow the reader to have a better understanding of what is the error of you suggested calibration method with respect to commercial devices. From this perspective, it would be useful that the error data that you provide in pixels in all your Tables would be expressed also in degrees of visual field.

- line 42: A SENSITIVE VARIABLE FOR AN ACCURATE CALIBRATION IS ALSO THE NUMBER OF CALIBRATION TARGETS AND THE CALIBRATED AREA. THERE ARE A COUPLE OF CITATIONS THAT WOULD BE WORT TO USE FOR THIS:

Canessa, A., Gibaldi, A., Chessa, M., Sabatini, S. P., & Solari, F. (2012). The perspective geometry of the eye: toward image-based eye-tracking. In Human-Centric Machine Vision. InTech

Gibaldi, Agostino, et al. "Evaluation of the Tobii EyeX Eye tracking controller and Matlab toolkit for research." Behavior research methods 49.3 (2017): 923-946.

Maiello, G., Harrison, W. J., & Bex, P. J. (2016). Monocular and binocular contributions to oculomotor plasticity. Scientific Reports, 6, 31861.

- In eye tracking data you will find samples relative to fixations, but also to saccades and smooth pursuits. The first category is useful for calibration, the next two usually are not. Do you do any distinctions in your algorithm?

- In Section 2.3 you might also consider another method. The transformation from raw gaze data of any kind to calibrated gaze data might be considered an affine transformation with some specific constraints, depending on what is the principle of the eye tracker. You might use the mapping algorithm used in the feature extraction algorithms for image recognition, like Scale-invariant Feature Transform (SIFT), which are generally open source or available in MATLAB. Also, they generally work very good with RANSAC, which you name in Sec. 2.4.

- RESULTS: It would be very nice to have for 1 subject some sample images where the PFT are highlighted, and with the fixation heatmap superimposed. This would allow a graphic description of the results, providing a decent assessment to the method you propose.

- RESULTS: The organization of the paper is quite messy. For instance, in lines 380-460 present more Methods than Results, and same for lines 481-495. Please, reorganize per paper in order that methods are described in the Methods section, results in the Results section, etc…

MINOR ISSUES

line 4: with user --> by the user

line 5: a person --> the user

line 5-6: The paper describes one of the possible methods --> This paper describes a possible method

line 8: then uses these --> then it uses these

line 24: utilizing eye tracking also possible --> utilizing eye tracking possible also

line 31: I GUESS THAT WHEN YOU SAY: 'It may be information about eye center coordinates" YOU ACTUALLY MEAN "It may be information about image coordinates of the pupil center". IF NOT, EXPLAIN.

line 36: IT IS A LITTLE MISLEADING WHEN YOU SAY THAT THE GAZE POINT IS IN SCENE COORDINATES. IF SO, WHAT IS THE REFERENCE FRAME OF THE SCENE? TO MY KNOWLEDGE, GAZE INFORMATION IS USUALLY PROVIDED IN TERMS OF

- SCREEN COORDINATES OF THE FIXATION POINT (X,Y) IN PIXEL

- ANGULAR COORDINATES OF THE GAZE DIRECTION (AZIMUTH,ELEVATION) IN DEGREES

- 3D COORDINATED OF THE FIXATION POINT (X,Y,Z) IN MILLIMITERS OR OTHERS

PLEASE EXPLAIN WHAT YOU MEAN.

line 38: time at the predefined locations on a scene --> time at predefined locations in the scene

OTHER WISE time at predefined locations on the screen

line 38-41: THIS SENTENCE IS NOT ENGLISH, PLEASE REPHRASE IT

line 47: gazing at not moving target is not natural. --> gazing at a static target is not a natural task

line 53: alternative ways of the calibration --> alternative calibration methods  

line 57-58: For this reason, a growing interest in methods able to perform the calibration implicitly, during normal user’s activities may be observed. --> For this reason, there is a growing interest in methods able to perform an implicit calibration during normal user’s activities.

line 64: explicit and active cooperation with users --> explicit and active cooperation of the user

line 68: about the scene, the user REMOVE THE COMMA

line 96: to use genuine human gazes --> to use gaze data

line 112: WHICH PAPER? I GUESS YOU MEAN "This paper describes..."

line 114: After finding this targets --> After finding these targets

line 114: THE Probable Fixation Target ARE THE CORE OF YOUR WORK, YOU BETTER NOT DEFINE THEN IN BRACKETS, BUT MAYBE DEDICATE A FULL SENTENCE TO THEM...

line 117: YOU ALREADY DEFINED THE PFT ACHRONIM...

line 132: WAHT IS THE ETCAL LIBRARY? DEFINE IT.

line 143: created by any of saliency models --> created by a saliency model

line 139-163: THE TEXT HERE IS VERY CHAOTIC. REPHRASE IT AND POSSIBLY PROVIDE A PSEUDO-CODE FOR YOUR ALGORITHM, SIMILARLY TO WHAT YOU DO FOR THE TABLE Algorithm 1.

line 209: THIS IS AN INTERESTING APPROACH, BUT IT IS OTHER THAN A GROUND TRUTH SALINECY MAP. PLEASE, CHANGE NAME TO SOMETHING MORE RELEVANT TO WHAT IT IS, LIKE "FIXATIONAL MAP" OR  "FIXATION SALIENCY MAP".

line 207-217: THESE TWO METHODS (GT AND EXP) SHOULD BE LISTED JUST LIKE THOTERS IN THE DOTTED LIST ABOVE.

line 270: THIS IS OTHER THAN CLEAR. YOU MAKE AN EXAMPLE AT LINE 279 WHICH IS MORE HELPFUL AND SHOULD BE MOVED HERE.

line 279: THIS CAN VARY AMONG DIFFERENT EYETRACKERS.

line 411-412: IS THIS THE NUMBER OF FIXATIONS PER IMAGE? PLEASE, EXPLAIN.

Author Response

We would like to thank both reviewers for their valuable comments. We corrected the paper according to their remarks, specifically:

·        We added a description of another experiment that calibrates users based on movie watching data (Sections 3.3 and 4.3)

·        We added a more comprehensive discussion about saliency model results – including significance analysis (in sections 4.1 and 4.2 and in Summary)

·        We corrected the structure of the paper by moving part of the Results section to a new Experiment section placed between Methods and Results sections.

·        We added some more introductory information about current technologies and calibration methods.

Additionally, we did our best to correct any language mistakes and the paper was checked by a native speaker.

All new text (apart from language corrections) is denoted with the blue color in the current text.

Specific comments/answers to reviews:

Reviewer 2:

IN GENERAL: language is not too bad, but it is still quite far from English. Please, have the paper checked by a native speaker.

We are sorry for all mistakes - the paper has been extensively corrected and read by a native speaker.

One big risk of your method is to incur in overfitting of the subject's data. Just like what is used in neural networks problems, it would be interesting to do a random division between the TRAINING gaze data (maybe 80% of the total or so) and the TEST gaze data (the remaining 20%).

It indeed may be a serious problem. However, in our case it does not exist because for testing purposes we used only gazes obtained during watching of reference screens. We additionally clarified it in the text (lines 469). 

INTRODUCTION and RESULTS: eye tracking is becoming very popular in the last years, it would help the clarity of the paper to provide a short introduction about the current technologies and frame them in the literature. I suggest these citations

Thank you for this suggestion, we incorporated the given citations in text (lines 18-22).

From this perspective, it would be useful that the error data that you provide in pixels in all your Tables would be expressed also in degrees of visual field.

We decided to use screen percent as our result because we felt that it has more intuitive meaning for people not involved in eye tracking research and working in HCI field. However, we added information how these values may be roughly recalculated to degrees of visual field (lines: 449-455, 474-480).

- line 42: A SENSITIVE VARIABLE FOR AN ACCURATE CALIBRATION IS ALSO THE NUMBER OF CALIBRATION TARGETS AND THE CALIBRATED AREA. THERE ARE A COUPLE OF CITATIONS THAT WOULD BE WORT TO USE FOR THIS:

Thank you for your suggestion, we added references to all the mentioned papers.

In eye tracking data you will find samples relative to fixations, but also to saccades and smooth pursuits. The first category is useful for calibration, the next two usually are not. Do you do any distinctions in your algorithm?

We are using only fixations in our algorithm. We added the more extensive description in the text to make it more obvious (lines 151-165).

In Section 2.3 you might also consider another method. The transformation from raw gaze data of any kind to calibrated gaze data might be considered an affine transformation with some specific constraints, depending on what is the principle of the eye tracker. You might use the mapping algorithm used in the feature extraction algorithms for image recognition, like Scale-invariant Feature Transform (SIFT), which are generally open source or available in MATLAB. Also, they generally work very good with RANSAC, which you name in Sec. 2.4.

This is a really good idea, somewhat similar to Alnajar et al. [24]. We used a different approach but the one described by the reviewer is worth further studies.

RESULTS: It would be very nice to have for 1 subject some sample images where the PFT are highlighted, and with the fixation heatmap superimposed. This would allow a graphic description of the results, providing a decent assessment to the method you propose.

Such an example as a movie was already presented (link in line 409). We additionally added a link to the new movie presenting the results of the third (movie watching) experiment (line 495). We hope that these movies will be incorporated into the final online version of the paper.

RESULTS: The organization of the paper is quite messy. For instance, in lines 380-460 present more Methods than Results, and same for lines 481-495. Please, reorganize per paper in order that methods are described in the Methods section, results in the Results section, etc…

We moved all parts of the previous Results session which involved more methods than results to a new Experiments session placed between Methods and Results (lines 379-500).

MINOR ISSUES

Thank you for all your language corrections, we applied them all and additionally a proof read was done by a native speaker.

line 31: I GUESS THAT WHEN YOU SAY: 'It may be information about eye center coordinates" YOU ACTUALLY MEAN "It may be information about image coordinates of the pupil center". IF NOT, EXPLAIN.

We corrected the sentence (line 37).

line 36: IT IS A LITTLE MISLEADING WHEN YOU SAY THAT THE GAZE POINT IS IN SCENE COORDINATES.

We rephrased this part of the text and hope it clearer now (lines 43-46).

line 114: THE Probable Fixation Target ARE THE CORE OF YOUR WORK, YOU BETTER NOT DEFINE THEN IN BRACKETS, BUT MAYBE DEDICATE A FULL SENTENCE TO THEM...

It was corrected (line 125).

line 132: WAHT IS THE ETCAL LIBRARY? DEFINE IT.

We added the explanation (line 144-145).

line 139-163: THE TEXT HERE IS VERY CHAOTIC. REPHRASE IT AND POSSIBLY PROVIDE A PSEUDO-CODE FOR YOUR ALGORITHM, SIMILARLY TO WHAT YOU DO FOR THE TABLEAlgorithm 1.

We rewrote the whole section and added a new schema that defines our algorithm, we hope it is more comprehensible now (lines 151-165).

line 209: THIS IS AN INTERESTING APPROACH, BUT IT IS OTHER THAN A GROUND TRUTH SALINECY MAP. PLEASE, CHANGE NAME TO SOMETHING MORE RELEVANT TO WHAT IT IS, LIKE "FIXATIONAL MAP" OR  "FIXATION SALIENCY MAP".

We changed the description (line 216).

line 207-217: THESE TWO METHODS (GT AND EXP) SHOULD BE LISTED JUST LIKE THOTERS IN THE DOTTED LIST ABOVE.

It has been changed (lines 216-220).

line 270: THIS IS OTHER THAN CLEAR. YOU MAKE AN EXAMPLE AT LINE 279 WHICH IS MORE HELPFUL AND SHOULD BE MOVED HERE.

The order of sentences and the structure of these paragraphs has been changed (lines 283-289).

line 279: THIS CAN VARY AMONG DIFFERENT EYETRACKERS.

The text has been changed to reflect specific properties of different eye trackers (lines 292-294).

Round 2

Reviewer 1 Report

The paper has improved since the previous submission. I did not have many remarks about the style and that has not changed. I am bit disappointed that there was no effort to shorten the exposition of the individual methods especially those that can easily be referenced to the ETRA paper for more details. The paper is now 5 pages longer. On the other hand, I appreciate that most of the extra text went to the discussion of the results.

My main concern was the novelty with respect to the ETRA paper [30]. This has been slightly eased by addition of the extended discussion and with the additional video experiment. That being said, I have a feeling that the video experiment has not been given a lot of care judging from how brief its exposition is and that the choice of the content is clearly not well suited for the saliency models. Overall, the amount of new content alone may be acceptable but I would still like to see a clear definition of what is cited from [30] and what is a new contribution. It should be clearly listed in the end of Section 1 (lines 139-145). Additionally, each of the Figures taken from [30] should have a citation in the caption (e.g. Figure 1, 5 and 7). Similarly, the linked Youtube videos also contain a description referring to the ETRA paper so I consider them citations of the previous work. In general, there should be citation every time a result from the previous paper is discussed.

Minor:

* There is still no reference from the text to the Table 3.